# SuperWeight Ensembles: Automated Compositional Parameter Sharing Across Diverse Architectures

## Abstract

Neural net ensembles boost task performance, but have excessive storage requirements. Recent work in efficient ensembling has made the memory cost more tractable by sharing learned parameters between ensemble members. Existing efficient ensembles have high predictive accuracy, but they are overly restrictive in two ways: 1) They constrain ensemble members to have the same architecture, limiting their usefulness in applications such as anytime inference, and 2) They reduce the parameter count for a small predictive performance penalty, but do not provide an easy way to trade-off parameter count for predictive performance without increasing inference time. In this paper, we propose SuperWeight Ensembles, an approach for architecture-agnostic parameter sharing. SuperWeight Ensembles share parameters between layers which have sufficiently similar computation, even if they have different shapes. This allows anytime prediction of *heterogeneous* ensembles by selecting a subset of members during inference, which is a flexibility not supported by prior work. In addition, SuperWeight Ensembles provide control over the total number of parameters used, allowing us to increase or decrease the number of parameters without changing model architecture. On the anytime prediction task, our method shows a consistent boost over prior work while allowing for more flexibility in architectures and efficient parameter sharing. SuperWeight Ensembles preserve the performance of prior work in the low-parameter regime, and even outperform fully-parameterized ensembles with 17% fewer parameters on CIFAR-100 and 50% fewer parameters on ImageNet.

## 1 Introduction

Ensembling aggregates the predictions of multiple models in an effort to boost task performance (Atwood et al., 2020; Ostyakov & Nikolenko, 2019) while also improving robustness, calibration, and accuracy (Lakshminarayanan et al., 2017; Dietterich, 2000). However, it also require more memory and compute since multiple models need to be trained, stored, and used for inference. Efficient ensembling reduces the total number of parameters by sharing parameters between members (*e.g.*, (Lee et al., 2015; Wen et al., 2020; Wenzel et al., 2020)). As illustrated in Figure 1(a), these methods share parameters while introducing diversity by a) perturbing the shared parameters to create distinct layer weights or b) perturbing layer inputs for each ensemble member. A significant drawback of these methods is they often make strong architectural assumptions, such as ensemble member homogeneity (*i.e.*, each member is the same architecture), which limits their use. For example, homogeneous ensembles are ill-suited to tasks like anytime prediction because one only has $n$ options for computational complexity, where $n$ is the number of ensemble members. In contrast, heterogeneous ensembles can select a subset of its ensemble members to provide a range of inference times (*e.g.*, a 4 member heterogeneous ensemble can adjust to $\binom{4}{1} + \binom{4}{2} + \binom{4}{3} + \binom{4}{4} = 15$ levels of inference latency). Thus, we present SuperWeight Ensembles, which allow for parameter-efficient anytime inference by using heterogeneous ensemble members. Figure 1(c) shows that our heterogeneous ensembles achieve state-of-the-art performance in anytime inference.

In this paper, we propose SuperWeight Ensembles, a method for efficient ensembling using automated parameter sharing between diverse architectures. A key challenge we address is learning where we can reuse parameters for heterogeneous ensembles, where ensemble members may have

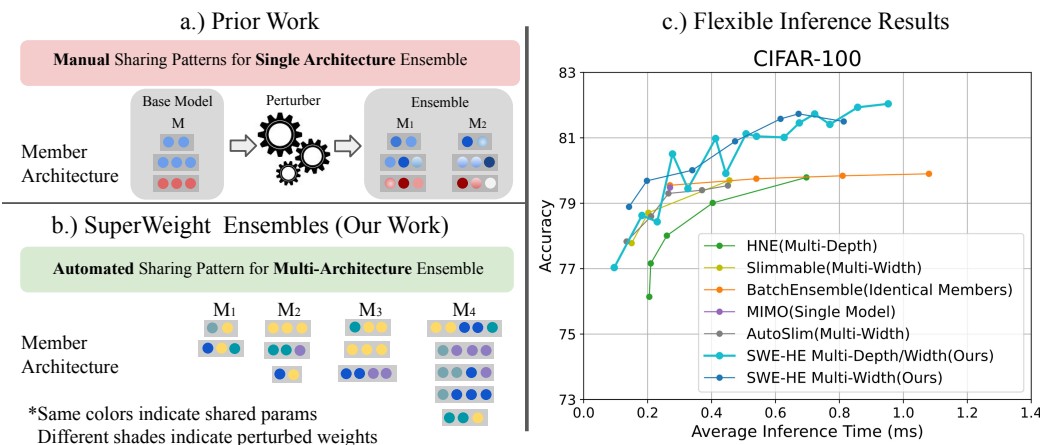

Figure 1: **Comparison to prior work. (a)** Prior work in efficient ensembling (*e.g.*, (Lee et al., 2015; Wen et al., 2020; Wenzel et al., 2020)) uses hand-crafted strategies that required ensemble members to have identical architectures and adds diversity by perturbing weights and/or features. In contrast, **(b)** illustrates our SuperWeight Ensembles, which learn effective soft parameter sharing between members, even for diverse architectures. As shown in **(c)**, this enables our approach to support a range of inference times while outperforming prior work in efficient ensembling and anytime inference (Ruiz & Verbeek, 2021; Havasi et al., 2021; Wen et al., 2020; Yu et al., 2019; Yu & Huang, 2019a) on CIFAR-100 using WRN-28-5 (Zagoruyko & Komodakis, 2016).

architectures that vary both the number of layers and/or channels. Prior work has explored strategies where parameters are shared between members that have constant depths, but varying widths (Yu & Huang, 2019b; Yu et al., 2019; Wang et al., 2020; Li et al., 2021), as well as cases where the widths are held constant, but the depths vary (Ruiz & Verbeek, 2021; Kaya et al., 2019; Huang et al., 2018; Yang et al., 2020; Wu et al., 2018). In contrast, as illustrated in Figure 1(b), SuperWeight Ensembles members can vary in both the widths and depths of the network. Our approach is built on the intuition that, despite being independent, two models trained on the same task likely have to detect similar features. However, these features may occur in different location(s) (*i.e.*, different layers) across ensemble members, especially when members represent distinct networks architectures. Our goal then is to detect where these recurring computations exist in our heterogeneous ensemble.

A neural network can be seen as a composition of feature detectors (Savarese & Maire, 2019). Motivated by this and the Hebbian principle that "neurons which wire together fire together", we propose a technique that groups detectors that frequently co-occur, which we call SuperWeights. Our SuperWeights are analogous to SuperPixels (Ren & Malik, 2003), which represent a semantically coherent region of pixels within an image, but where our SuperWeights represent feature detectors which frequently co-occur. An example of a SuperWeight would be filters of a CNN that capture a unique pattern. Prior work has shown these patterns may repeat across many types of classification networks (Raghu et al., 2021), which we take advantage of in our approach. A single layer may require a concatenation of several SuperWeights, each of which may be shared with layers within the same ensemble member or across members. To learn these complex sharing patterns, we propose a gradient analysis approach for determining where SuperWeights can be reused.

To further improve our model's parameter efficiency, we also take advantage of template mixing (*e.g.*, Bagherinezhad et al. (2017); Plummer et al. (2022); Savarese & Maire (2019)). We construct SuperWeights using a weighted linear combination of templates made up of trainable parameters which we call Weight Templates. This creates a hierarchical representation for neural network weight generation, where we begin by combining Weight Templates to create SuperWeights, then concatenating together SuperWeights to create the layer weights used by each ensemble member. Two SuperWeights using the same Weight Templates may learn different linear coefficients for combining templates, allowing for per-layer SuperWeight sub-specialization. This template mixing can boost performance when reusing the same parameters many times, as each combination of templates can use a unique set of coefficients, and it also helps us support a wide range of parameter budgets. Adjusting the parameter count can be accomplished by increasing or decreasing the number of shared templates, allowing for parameter budget vs performance trade offs with no change in

architecture. In contrast, prior work in efficient ensembling (*e.g.* Lee et al. (2015); Wen et al. (2020); Wenzel et al. (2020)) can only support a single parameter budget per architecture. Our contributions are as follows:

- We propose SuperWeight Ensembles, a new technique for automated parameter sharing across layers and ensemble members, even when members have different architectures.
- We demonstrate that using our approach with *homogeneous* architectures matches performance of prior parameter efficient ensembles on CIFAR-10 (Krizhevsky et al., 2009), CIFAR-100 (Krizhevsky et al., 2009), CIFAR-100-C (Hendrycks & Dietterich, 2019), and ImageNet (Deng et al., 2009) in the low parameter budget setting.
- We also illustrate SuperWeight Ensembles' ability to tradeoff between its parameter budget and performance to surpass fully-parameterized ensembles while still using 17% fewer parameters on CIFAR-100 and 50% fewer parameters on ImageNet.
- We show that using our approach with *heterogeneous* architectures achieves state-of-the-art anytime inference results on CIFAR-10 and CIFAR-100 by evaluating a subset of ensemble members for different inference budgets, an application not supported by prior work in efficient ensembling.

## 2 RELATED WORK

Ensembles have been shown to improve predictive accuracy and uncertainty estimates of classification models (Dietterich, 2000; Lakshminarayanan et al., 2017). Prior work has shown this is due, in part, to the fact that different neural network initializations result in diverse final solutions (Fort et al., 2019), which helps to decorrelate errors and boost performance. In fact, Wasay & Idreos (2021) show that ensembling many smaller models sometimes is a more efficient way to use additional compute than making a single model larger. Ensembles' major drawback is that they are memory and compute intensive since multiple models must be trained and used for inference. To address this, many **efficient ensembling** methods share parameters between members (*e.g.*, (Havasi et al., 2021; Lee et al., 2015; Teterwak, 2014; Wen et al., 2020)), either by introducing a branching point between shared and unshared layers (Lee et al., 2015; Teterwak, 2014) or, more recently, perturbing the input features and/or shared parameters to create ensemble members (Gal & Ghahramani, 2016; Havasi et al., 2021; Wen et al., 2020). However, as mentioned in the introduction, these methods use hand-crafted strategies that rely on each ensemble member having the same architecture. Additionally, they require architecture changes in order to alter parameter budgets since the parameter sharing pattern is fixed. In contrast, our SuperWeight Ensembles automatically learn how to share parameters between multiple architectures given a parameter budget, enabling us to generalize to more tasks and settings like anytime inference.

In **anytime inference** the goal is to make a high performing prediction within a given inference time budget. To address this task, most methods learn to make predictions using subnetworks that have varying computational requirements to support a range of inference times (*e.g.*, (Bolukbasi et al., 2017; Iuzzolino et al., 2021; Ruiz & Verbeek, 2021; Wu et al., 2018; Yu & Huang, 2019b; Yu et al., 2019; Huang et al., 2018)). These subnetworks vary either the network width (Yu & Huang, 2019b; Yu et al., 2019) or network depth (Bolukbasi et al., 2017; Iuzzolino et al., 2021; Ruiz & Verbeek, 2021; Wu et al., 2018; Huang et al., 2018). In contrast, SuperWeight Ensembles learn heterogeneous ensembles that can vary both width and depth, or members could use a completely different network architecture altogether, increasing the diversity and inference times we can support. In addition, unlike in efficient ensembling, where individual models may perform poorly but still obtain good performance since they have high diversity across the set of models, in anytime inference the performance of individual models hold greater weight since they are meant to support a specific inference time budget. SuperWeight Ensembles obtain state-of-the-art results in both.

**Neural Architecture Search (NAS)** is another approach for finding high performance networks. Neural Ensemble Search (Zaidi et al., 2020) searches for optimal ensemble member architectures by evolving ensemble members. Single-shot NAS methods search subnetworks over a pre-trained supernetwork (Yu & Huang, 2019a; Yu et al., 2020a). However, strict parameter sharing over child sub-networks is suboptimal. Therefore, few-shot NAS (Zhou et al., 2022; Hu et al., 2022) has been introduced, which train several supernetworks. GM-NAS (Hu et al., 2022) uses a gradient-based splitting criterion to find supernetworks, and CLOSE (Zhou et al., 2022) decouples parameter assignment from operations. Nevertheless, NAS is very costly due to a search over architectures.

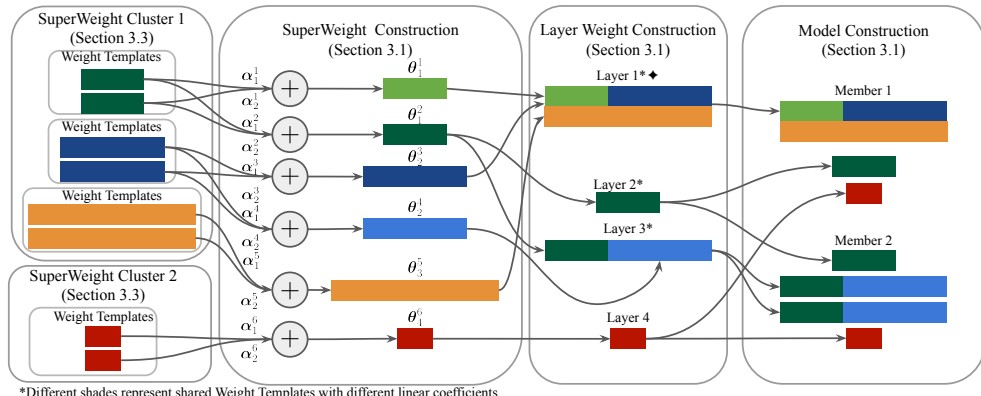

Figure 2: **SuperWeight Ensembles overview. Left:** Weight Templates are clustered into Super-Weight Clusters; layers can only share SuperWeights from Weight Templates in the same cluster (Section 3.2.2) **Center Left:** SuperWeights are constructed from linear combinations of Weight Templates (see Eq. (1)) using the SuperWeight Clusters learned in Section 3.3. **Center Right:** The SuperWeights assigned to a layer are then concatenated to generate layer weights. We automatically identify when we should use the same linear coefficients or different coefficients for generating a layer's SuperWeight using the procedure in Section 3.2.2. **Right:** These layer weights are then assembled into ensemble members, which can have diverse architectures.

# 3 SuperWeight Ensembles

Given a parameter budget $B$ and a set of target model architectures $M$, one for each ensemble member, our goal is to generate the weights of the layers of each member that maximize task performance. To address this task, we propose SuperWeight Ensembles, an automated approach for sharing parameters between ensemble members, even when they contain diverse architectures (*i.e.*, ensemble members may vary in both the widths and depths from each other, or can be different architecture families). Our approach handles this challenge by constructing layers from one or more SuperWeights, which are linear combination of templates and get reused by multiple layers and capture a single operation on the input features (*e.g.*, edge or texture detectors). Then, to generate the weights for a single layer, we must first construct SuperWeights from the trainable parameters held by Weight Templates (discussed in Section 3.1), and then *concatenate* together all SuperWeights used by the layer to create its final weights (process illustrated in center-right column of Figure 2).

This hierarchy provides a very parameter efficient representation, as the trainable parameters used by each Weight Template may be shared by multiple SuperWeights, while still being diverse as different coefficients are used to combine the Weight Templates into the SuperWeights. The SuperWeights themselves can also be shared by one or more layers, either from the same ensemble member, or different ensemble members. However, some layers cannot be shared without substantial performance degradation. Therefore, we learn where parameter sharing is most advantageous by starting with strict sharing where many layers share parameters, then removing sharing where it hurts performance. The overall algorithm consists of the following steps:

1. We start by determining the sizes of the SuperWeights needed for the network (Section 3.2.1), and construct an initial sharing pattern based on relative depth.
2. After a small number of training epochs, we refine this sharing pattern using gradient analysis to determine which layers should share Weight Templates, resulting in the creation of our SuperWeight Clusters (Section 3.3).
3. We reinitialize the entire network using our SuperWeight Clusters and learn which Weight Template Coefficients $\alpha$ (*i.e.*, which SuperWeights) should be shared across layers (Section 3.2.2).
4. We resume training until we achieve our stopping criteria *e.g.* number of training epochs.

## 3.1 Generating SuperWeights from Templates

Our ensemble uses a set of Weight Templates as a fundamental building block for transforming shared parameters into layer-specific weights – the SuperWeight. Traditional methods of (hard)

parameter sharing directly reuse weights (*e.g.*, (Collobert & Weston, 2008; Havasi et al., 2021; Gal & Ghahramani, 2016; Zhang et al., 2014)), but this limits ensemble diversity as shared layers then encode the same learned function. Instead, we take inspiration from recent work in cross-layer parameter sharing (*e.g.*, (Bagherinezhad et al., 2017; Plummer et al., 2022; Savarese & Maire, 2019)) that performs template mixing. In these works, parameters and layer weights are decoupled; each layer's weights are a linear combination of parameter matrices called templates. We build each SuperWeight in our SuperWeight ensembles in the same way. More formally, let us consider generating SuperWeight $\theta_g^k$, where $k$ refers to SuperWeight $k$ generated using Weight Templates $g$. $g$ contains a set of templates $T_{1,\dots,N}^g$ with the same dimensions as the SuperWeight $\theta_g^k$, which is generated via a linear combination using coefficients $\alpha^k$, *i.e.*,

$$\theta_g^k = \sum_{i=1}^{N} \alpha_i^k T_i^g \tag{1}$$

Multiple SuperWeights can share $g$ while having different coefficients $\alpha^k$. To create a layer's weights, one or more SuperWeights are generated using Eq. (1) and then concatenated together (details in Section 3.2.1). Templates and coefficients are learned jointly via gradient descent.

Plummer et al. (2022) also shared parameters via template mixing between layers of different sizes, but they would directly reshape the available parameters into templates of the target layer size. This means that it would be very difficult to accurately represent subnetworks within a larger network. For example, let us assume we are generating the weights for two networks that are identical except that network $X$ is twice as wide as network $Y$. Using the approach from Plummer et al. (2022), we would either have to generate layers for $X$ and $Y$ independently, or we would generate the layers of $X$, and then slice out the weights that would fit network $Y$. We found that the first option was difficult to optimize that resulted in lower performance in our experiments. The second option would reduce diversity, as network $X$ would contain exactly the same weights as network $Y$. Instead, our SuperWeight Ensembles can directly optimize any subnetworks, like in the second option in our example above, but while minimizing any diversity loss (details in the next section).

## 3.2 Constructing Networks from SuperWeights

### 3.2.1 Determining SuperWeight Sizes

The target sizes for our SuperWeights are based on layer shapes. Consider the case where we are generating weights for a set of layers $\ell_{1,\dots,S}$, where each layer is a different size. We create an initial set of SuperWeights by sorting the layers in order from smallest to largest. The first SuperWeight we would create would be the same size as the smallest layer $\ell_i$, and every layer within that SuperWeight Cluster that was the same size as $\ell_i$ would initially share the same SuperWeight. Then, when we generate the weights for the next layer $\ell_{i+1}$, we assume we have the SuperWeight from layer $\ell_i$. Thus, the SuperWeights for layer $\ell_{i+1}$ need only generate the additional weights required beyond those provided by the SuperWeights from $\ell_i$. For example, if $\ell_i$ needed 100K parameters and $\ell_{i+1}$ needed 400K, then the new SuperWeights would generate $400 - 100 = 300$K weights. If $\ell_{i+1}$ is larger than $\ell_i$ in only a single dimension, then only a single new SuperWeight is generated. If it's larger in both input and output channels, $\ell_{i+1}$ generates two new SuperWeights. The first is to extend the size in a input channels dimension, and the second is concatenated in the output channel dimension. See Layer 1 in the center-right column of Figure 2 for a graphical illustration. Any subsequent layers would follow the same process, but would have all the weights from the previous layers (*i.e.*, $\ell_{i+2}$ would have 400K weights from $\ell_{i+1}$, 100K of which originally came from $\ell_i$).

### 3.2.2 Learning when SuperWeights should share Weight Template coefficients

We use the initial set of SuperWeights from Section 3.3 for the first $E$ epochs of training (we found $E = 10$ worked well). After that, we analyze which layers that share the same SuperWeight benefit from decoupling linear coefficients combining Weight Templates. Similar to recent multi-task work (Yu et al., 2020b), we compute layers which have conflicting gradients of the loss with respect to linear coefficients $\alpha$. These layers are playing a gradient tug-of-war, hurting optimization. In other words, if the sum of gradients from two layers is close to zero, no learning occurs. To prevent this, we cluster layers by gradient similarity. We place the layers sharing the same SuperWeight

into a priority queue by the cosine similarity with respect to the gradients of the shared coefficients aggregated over an epoch. We pop the first two layers, $\ell_i, \ell_j$ off the queue and checking whether their cosine similarity exceeds some threshold $\beta$, *i.e.*,

$$\psi_{coef.} = \cos\left(\frac{\partial\mathcal{L}}{\partial\alpha^{k,j}}, \frac{\partial\mathcal{L}}{\partial\alpha^{k,i}}\right) > \beta, \tag{2}$$

where $\alpha^{k,i}$ is the shared coefficient corresponding to layer $\ell_i$, and $\mathcal{L}$ is the loss. If both layers belong to an existing group and they satisfy Eq. (2), we merge their groups, meaning that the layers within a group share SuperWeights. If pair of layers satisfy Eq. (2), but only one layer is in an existing group, then we add the new layer into that group's set. Once we have a pair of layers that does not satisfy Eq. (2), we split any ungrouped layers into their own individual groups (*e.g.*, 4 ungrouped layers would result in 4 additional single layer groups). We set $\beta$ via grid search on a validation set. See Appendix B for pseudocode for our priority-queue assignment procedure. After we have identified the groups of similar layers, each obtains their own copy of the coefficients for that SuperWeight $\alpha^k$ and we resume training. This creates a new SuperWeight for each group of layers whose gradients point in a similar direction, allowing for layer specialization.

### 3.3 LEARNED PARAMETER SHARING WITH SUPERWEIGHT CLUSTERS

In Section 3.2.2 we describe a procedure for decoupling linear coefficients, softening the sharing between layers. However, some layers may not find any sharing beneficial. To account for such cases, we first separate layers into SuperWeight Clusters, sets of layers where sharing is most beneficial, and do not share parameters between layers in different clusters. Plummer et al. (2022) automatically learned where parameter sharing could be effective by forcing layers to share templates, but not coefficients, and then used $K$-means clustering on the coefficients to group layers that produced similar weights (due to having similar coefficients). This has two major drawbacks. First, layers are clustered based on the final value of the coefficients, but two layers may be drifting apart, and waiting until convergence is also not optimal due to the resulting increase in training time. Second, sharing parameters across all layers requires upsampling/downsampling templates for layers needing different numbers of weights, which is accomplished using bilinear interpolation. This effectively warps the receptive field of the parameters, so they were not truly comparable across layers, while also being computationally expensive to do every layer in a forward pass.

Our SuperWeight Ensembles are designed to address these issues in prior work. Specifically, since we break layers into combinations of SuperWeights, they do not need to be upsampled/downsampled since the larger layer would simply share a subset of its weights with the smaller layer. In addition, instead of each layer that shares a SuperWeight getting their own coefficients and then clustering them (effectively adapting the approach of Plummer et al. (2022) to use our SuperWeights, which we compare to in Table 3), we force all layers to share coefficients so that we can tell if two layers should share the SuperWeight coefficients by analyzing their gradients.

More formally, given a set of layers $\ell_{1,\ldots,S}$ in our ensemble, our goal is to determine which layers can share parameters. We begin by creating a set of initial SuperWeight Clusters that we will refine. First, we split layers $\ell_{1,\ldots,S}$ into groups by operation (*e.g.*, convolutional and fully connected layers get split into different clusters). When instantiating convolutional layers, we consider layers with different kernel sizes as performing different operations, so convolutional layers with $1 \times 1$ and $3 \times 3$ kernels never belong to the same group. Since shallow and deep layers can be assumed to perform entirely different types of computation, we create $N$ bins based on the depth of a layer and ensure layers that are in different depth bins are in different groups. Layers from different members are placed in the same bin; ensuring cross-member parameter sharing.

Once we obtain the initial layer groupings using the process above, we create a set of shared Super-Weights using the procedure from Section 3.2.1, independently per group. The set of SuperWeights belonging to a single layer grouping is called a *SuperWeight Cluster*. At this point, every set of Weight Templates has exactly one set of coefficients. For example, if a cluster has three layers that all need a SuperWeight of 100K dimensions, then each layer would use a shared Weight Template that produces the same 100K dimension SuperWeight. Thus, we can infer that if the gradients of the loss w.r.t. the layers sharing SuperWeights are misaligned, then the model will have optimization difficulties. To capture this, we modify the procedure in Section 3.2.2. While Section 3.2.2 discussed gradient conflicts between coefficients, in this case the concern is with overly restrictive

Table 1: **Homogeneous ensembling comparison** on CIFAR-100 (clean) (Krizhevsky et al., 2009) CIFAR-100-C (corrupt) (Hendrycks & Dietterich, 2019), and CIFAR-10 using WideResNets (Zagoruyko & Komodakis, 2016) averaged over three runs. **(a)** shows that our approach outperforms prior work in efficient ensembling (Wen et al., 2020; Havasi et al., 2021). **(c)** compares the performance of increasing the number of parameters (without changing the architecture) using our approach compared to Deep Ensembles, which trains 4 independent networks as members.

| | | CIFAR-100 (clean) | | | CIFAR-100-C | | | CIFAR-10 | | |
| | Method | Params | Top-1 ↑ | NLL ↓ | ECE ↓ | Top-1 ↑ | NLL ↓ | ECE ↓ | Top-1 ↑ | NLL ↓ | ECE ↓ |
|---|---|---|---|---|---|---|---|---|---|---|---|
| **(a)** | WRN-28-10 | 36.5M | 79.8 | 0.875 | 8.6 | 51.4 | 2.70 | 23.9 | 96.0 | 0.159 | 2.3 |
| | BE | 36.5M | 81.5 | 0.740 | 5.6 | **54.1** | 2.49 | 19.1 | 96.2 | 0.143 | 2.1 |
| | BE + EnsBN | 36.5M | 81.9 | n/a | 2.8 | **54.1** | n/a | 19.1 | 96.2 | n/a | 1.8 |
| | MIMO | 36.5M | 82.0 | 0.690 | 2.2 | 53.7 | 2.28 | 12.9 | 96.4 | 0.123 | 1.0 |
| | Thin Deep Ensembles | 36.5M | 81.5 | 0.694 | **1.7** | 53.7 | 2.19 | 11.1 | 96.3 | **0.115** | 0.8 |
| | SWE-HO (**Ours**) | 36.5M | 82.2 | 0.702 | 2.7 | 52.9 | **2.17** | 10.3 | 96.3 | 0.120 | **0.8** |
| | SWE-HE (**Ours**) | 36.5M | **82.4** | **0.663** | 3.0 | 53.0 | **2.17** | **10.0** | **96.5** | 0.115 | **0.8** |
| **(b)** | Deep Ensembles | 146M | 82.7 | **0.666** | **2.1** | 54.1 | 2.27 | 13.8 | **96.6** | **0.114** | 1.0 |
| | SWE-HO (**Ours**) | 120M | **82.9** | **0.666** | 2.2 | **54.7** | **2.00** | **10.3** | **96.6** | 0.119 | **0.8** |

sharing of Weight Templates. Thus, we compute gradient similarity over SuperWeights rather than coefficents. However, some layers may be composed of multiple SuperWeights, only some of which may share with other layers in the initial cluster. For example, one layer $i$ may be composed of two SuperWeights, while another layer $j$ may use three (two of which are shared with layer $i$). $v_{i,j}$ refers to the set of SuperWeights shared by layers $i$ and $j$. $W_i$ refers to the instantiated weights of layer $i$. Then the gradient similarity between $W_i$ and $W_j$ would be computed as:

$$\psi_{SW} = \cos\left(\frac{\partial \mathcal{L}}{\partial W_i}\frac{\partial W_i}{\partial v_{i,j}}, \frac{\partial \mathcal{L}}{\partial W_j}\frac{\partial W_j}{\partial v_{i,j}}\right) > \tau, \quad (3)$$

where $\tau$ is a hyperparameter representing the minimum threshold for which two layers that share $v_{i,j}$ should remain in the same cluster. We found $\tau = 0.1$ worked well in our experiments. We first train our model for 20 epochs (10% of the training time in our experiments) using our initial SuperWeight Clusters (step 1 of training algorithm at the end of Section 3). Then we create our refined SuperWeight Clusters by using Eq. (3) as a replacement for Eq. (2) for the priority-queue based clustering from Section 3.2.2 (step 2 of our algorithm). Following Plummer et al. (2022), we reinitialize our ensemble and re-train from scratch with our new SuperWeight Clusters. Finally, after some more training, we decouple linear coefficients (step 3 of our algorithm, which is described in Section 3.2.2), and continue training (step 4 of our algorithm).

## 4 EFFICIENT ENSEMBLING EXPERIMENTS

We first compare our method to other efficient ensembling methods in a common application scenario; where all ensemble members share parameters but have the same architecture. In this special case; each layer consists of only a single SuperWeight. Implementation details are in the Appendix. **Datasets.** We evaluate our method on three standard benchmarks: CIFAR-10 (Krizhevsky et al., 2009), CIFAR-100 (Krizhevsky et al., 2009), and ImageNet (Deng et al., 2009). Additionally, as an advantage of ensembles is their robustness to out-of-distribution data, we evaluate the robustness of our method on CIFAR-100-C (Hendrycks & Dietterich, 2019), which is a version of the CIFAR-100 test set which is corrupted with distortions such as Gaussian blur and JPEG compression. **Metrics.** For a given parameter budget, we compute Top-1 accuracy in addition to calibration metrics (Guo et al., 2017): Negative Log-Likelihood (NLL) and Expected Calibration Error (ECE).

**Results.** Table 1(a) compares our SuperWeight Ensembles (SWE-HO) on the CIFAR-100 CIFAR-100-C, and CIFAR-10 with prior work in efficient ensembling. Note that all the methods boost performance over a single model without requiring additional model parameters. However, our SuperWeight Ensembles outperforms all other methods on CIFAR-100 when using 36.5M parameters. Unlike methods like BatchEnsemble (BE) (Wen et al., 2020) and MIMO (Havasi et al., 2021), which

Table 2: **Homogeneous ensembling comparison** on ImageNet using ResNet-50. **(a)** shows that our approach outperforms or is on par with prior work in efficient ensembling. **(b)** increases the number of parameters (without changing the architecture) using our approach compared to Deep Ensembles. See Section 4 for discussion

| | Method | Params | Top-1 |
|---|---|---|---|
| **(a)** | ResNet-50 (He et al., 2016) | 25.6M | 76.4 |
| | BE (Wen et al., 2020) | 25.6M | 76.7 |
| | MIMO (Havasi et al., 2021) | 25.6M | **77.5** |
| | SWE-HO (**Ours**) | 25.6M | 77.2 |
| **(b)** | Deep Ensembles | 102.4M | 77.5 |
| | SWE-HO (**Ours**) | 51.2M | **77.9** |

cannot change the number of parameters without making architecture adjustments to the widths and/or number of layers, our SuperWeight Ensembles can support any parameter budget without requiring architecture changes by adjust the number of templates or the amount of sharing between layers. Thus, if the number of parameters are not a concern, one can simply provide our approach with more parameters to boost performance as illustrated in Table 1(c), where our approach can outperform standard Deep Ensembles, which trains completely independent networks for ensemble members, while still retaining 17% fewer parameters. Table 1(a) also shows that heterogeneous SuperWeight Ensembles (SWE-HE), consisting of a WRN-34-8, 28-12, 28-10, and 28-8, outperforms SWE-HO in over half of the metrics while also supporting many inference times. A similar pattern is reported in Table 2 on ImageNet. Specifically, Table 2(a) shows we obtain on par or better performance than prior work, demonstrating that our approach can generalize to other settings including large scale recognition tasks. Table 2(b) also reports better performance by boosting parameter budgets, where we outperform standard Deep Ensembles with 50% fewer parameters. We do not include MIMO and BE in Table 1(b) and Table 2(b) as increasing number of parameters increases training and inference time, due to necessary increased width or depth of networks. This is not the case in SuperWeight Ensembles. In addition, methods like MIMO and BE cannot learn heterogeneous ensembles for anytime inference (as we do with SuperWeight Ensembles in Section 4.1), since their approaches rely on their ensemble members all being the same architecture.

## 4.1 ANYTIME INFERENCE EXPERIMENTS

In anytime inference the goal is to make high-performing predictions within a given time budget. As the time budget increases, a good method will use the additional time to improve performance. A homogeneous ensemble, like those used in the experiments in Section 4, would only provide limited time budgets as each member has an identical computational complexity. Thus, their flexibility is limited because one has only $M$ options for computational complexity, where $M$ is the number of ensemble members. However, for a heterogeneous ensemble this limitation is removed with an ensemble since each member provides a different inference time resulting in $\binom{M}{1} + \binom{M}{2} + ... + \binom{M}{M}$ possible inference times to choose from. This results in a highly effective anytime inference model. We note that although we evaluate our individual ensemble members in series, our method is trivial to parallelize to increase inference speeds by using multiple GPUs. This is unlike many early-exit anytime inference methods (*e.g.*, (Bolukbasi et al., 2017; Huang et al., 2018; Kaya et al., 2019; Li et al., 2019; Teerapittayanon et al., 2016; Yang et al., 2020)), which are intrinsically serial. We use two settings for our Heterogeneous SuperWeight Ensembles (SWE-HE) in our experiments: SWE-HE -Multi-Width, which trains a three member WRN (Zagoruyko & Komodakis, 2016) ensemble WRN-28-[7,4,3], and SWE-HE-Multi-Depth/Width, a four member ensemble WRN-28-[7,4] 16-[7,4]. SSNs (Plummer et al., 2022) and Slimmable (Yu et al., 2019) use the same ensemble configurations, but other approaches use method-specific strategies (*e.g.*, HNE (Ruiz & Verbeek, 2021) trains a set of tree-nested ensembles). See Appendix A for additional details.

**Results.** Figure 3 reports top-1 accuracy vs. average inference time using a single P100 GPU on CIFAR-10 and CIFAR-100 (Krizhevsky et al., 2009). When comparing to other dynamic width methods such as Slimmable (Yu et al., 2019), Universally Slimmable (Yu & Huang, 2019b) and AutoSlim (Yu & Huang, 2019a) models, SuperWeight Ensembles perform on par or better than them for inference times they support, but our approach can provide a wider range of inference times that improve performance. We reiterate that other efficient ensembling methods such as BatchEnsemble (Wen et al., 2020) and MIMO (Havasi et al., 2021) are not suitable for Anytime Inference, because each ensemble member has the exact same inference time. This results in a very limited set of possible inference times (see Figure 1). We also significantly outperform the tree-based ensemble

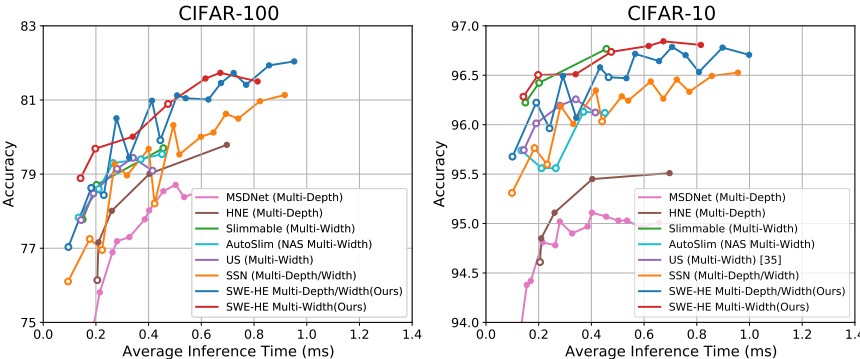

Figure 3: **Anytime Inference comparison** using top-1 accuracy vs. inference time (ms) averaged over three runs using a single P100 GPU on CIFAR-100 and CIFAR-10 (Krizhevsky et al., 2009) with WRN backbones (Zagoruyko & Komodakis, 2016) for most methods (Huang et al., 2018; Plummer et al., 2022; Yu & Huang, 2019a; Yu et al., 2019; Yu & Huang, 2019b), except HNE (Ruiz & Verbeek, 2021) which modifies ResNet-50 (He et al., 2016).

Table 3: **Multi-width SuperWeight Cluster creation comparison** using top-1 accuracy on CIFAR-100 averaged over three runs. See Section 4.1 for discussion

| Method | WRN-28-3 | WRN-28-4 | WRN-28-7 | Full Ensemble |
|---|---|---|---|---|
| Shared Coefficients | $77.3 \pm 0.09$ | $78.0 \pm 0.15$ | $79.7 \pm 0.10$ | $80.4 \pm 0.06$ |
| Single SuperWeight Cluster | $76.3 \pm 0.37$ | $77.6 \pm 0.30$ | $78.8 \pm 0.20$ | $81.1 \pm 0.17$ |
| Depth-binning | $76.8 \pm 0.15$ | $77.7 \pm 0.17$ | $79.2 \pm 0.19$ | $81.4 \pm 0.15$ |
| Coefficient Clustering (Plummer et al., 2022) | $76.1 \pm 0.19$ | $77.0 \pm 0.15$ | $78.9 \pm 0.15$ | $80.7 \pm 0.14$ |
| SWE-HE (**Ours**) | $\mathbf{78.9 \pm 0.26}$ | $\mathbf{79.7 \pm 0.08}$ | $\mathbf{80.9 \pm 0.03}$ | $\mathbf{81.5 \pm 0.13}$ |

HNE (Ruiz & Verbeek, 2021) on both datasets, as well as the early-exit model MSDNet Huang et al. (2018). Lastly, we use SuperWeight Ensembles to construct a dynamic width and depth network. Our main comparison is to adaptation of Shapeshifter Networks Plummer et al. (2022) to ensemble dynamic widths and depths. We show a consistent improvement over Shapeshifter Networks across inference times. These results demonstrate that SuperWeight Ensembles can share parameters across members of diverse architectures more effectively than other approaches.

**Comparison of SuperWeight Clustering methods.** Table 3 demonstrates the effectiveness of our SuperWeight Clustering approach described in Section 3.3. We provide four baselines: *Shared Coefficients*, which learns SuperWeight Clusters, but shares coefficients between all layers (*i.e.*, removing Section 3.2.2); *Single SuperWeight Cluster*, which allows layers to have their own coefficients, but does not learn clusters (*i.e.*, removing Section 3.3); *Depth-binning*, a heuristic used for our initial SuperWeight Clusters in Section 3.3; and *Coefficient Clustering* (Plummer et al., 2022), which clusters coefficients $\alpha$ in Eq. (1) to group layers. Our results show that our approach outperforms all of our baselines. Notably, we find we that Coefficient Clustering performs in par or worse than other baselines. In contrast, our gradient analysis approach (Section 3.3) takes into account the direction of change rather than just the current value of the coefficients. Thus, we obtain a 2% gain on individual models and a small boost to ensembling performance with our approach (Table 3). We show a visualization of SuperWeight cluster assignment in Appendix C.6.

## 5 CONCLUSION

We introduce SuperWeight Ensembles, a method for parameter sharing in heterogeneous ensembles. SuperWeight Ensembles outperform existing anytime prediction work by leveraging gradient information for parameter sharing. Our automatic sharing improves single member performance by 2% compared to the baselines. SuperWeight Ensembles also match performance of efficient ensembles in the low-parameter regime, compared to prior work. When we add parameters, we outperform even deep ensembles on ImageNet with 50% of the parameters. We believe that SuperWeight Ensembles are a promising step forward in parameter-efficiency. Future work will include more deeply exploring architecture diversity; Gontijo-Lopes et al. (2021) show that model architecture heterogeneity can be a key contributor to ensemble diversity on challenging tasks.

## 6 ETHICS STATEMENT

Ensembles are more robust and better calibrated than single models, which can translate directly to fairer and safer deployments. Nevertheless, we caution readers that even more robust and accurate models should not be trusted blindly. It is prudent to thoroughly audit all models before and during deployment, and consider how they may affect the lives of others.

Similarly, effective anytime inference models allows one to use less compute, potentially running networks in efficient modes and conserving energy. However, they can also be used to maximize the use of compute if it's available, using *more* energy with corresponding drawback. We urge readers to be aware of the carbon and energy footprint of the models they train.

## 7 REPRODUCIBILITY STATEMENT

We provide all training details in the Appendix, and provide code in the Supplementary Materials. At acceptance, we will provide GitHub repository for readers.

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

## A  IMPLEMENTATION DETAILS

For the readers convenience, we review the training details of both CIFAR Krizhevsky et al. (2009) and ImageNet (Deng et al., 2009) experiments. Note these details were used for both the efficient ensembling (Section 4) and the anytime inference results (Section 4.1).

**CIFAR:** For CIFAR-10 and CIFAR-100 experiments, we train for 200 epochs with an initial learning rate of 0.1. We use SGD with a Nestorov momentum value of 0.9. We use a weight decay value of $5e-4$ on all parameters except weight coefficients. We decay the learning rate by a factor of 0.2 at 60, 120,and 180 epochs. We use a batch size of 128, and use asynchronous BatchNorm across two devices (so BatchNorm batch size is 64). We pad images by 4 pixels and crop to 32 x 32 pixels, and also randomly flip and normalize such that it is zero-mean for training. We use the WideResNet (Zagoruyko & Komodakis, 2016) architecture for all CIFAR experiments.

**ImageNet:** For ImageNet experiments we use the ResNet-50 architecture (He et al., 2016). We train for 90 epochs with a learning rate of 1.6. We use SGD with non-Nestorov momentum value of 0.9. We use a batch size of 1024. We decay the learning rate at epochs 30, 60, and 80 by a factor of 0.1. We use a label smoothing value of 0.1, and a weight decay of 0.0001. The weight decay is not applied to batch norm parameters. We do a linear warmup for the first 10 epochs. We use standard Inception-style data augmentation. For ImageNet experiments we use 64 16GB V100 GPUs.

### A.1  EFFICIENT ENSEMBLE EXPERIMENTS

**SuperWeight Ensembles:** We use WRN-28-10 models for all of our efficient ensembling experiments. For our Efficient Ensemble models, we found that each layer having it's own SuperWeight works well. Each SuperWeight is constructed from 5 templates. Following Plummer et al. (2022), templates are allocated by iterating over the learnable parameters in a round-robin fashion over layers of the same shape. This is done with a double loop, with the outer loop over ensemble members and the inner loop over layers, to make it easy to modulate parameter count. In order to share parameters between functionally similar layers in CIFAR experiments, corresponding layers across ensemble members explicitly share templates. Because all members share the same architecture, we find doing gradient-based SuperWeight Cluster learning is not needed.

### A.2  ANYTIME INFERENCE EXPERIMENTS

**SuperWeight Ensembles:** We train all SuperWeight Ensembles on CIFAR-10 and CIFAR-100 using the settings above except we train on a single GPU, the learning rate is decayed at epochs 60, 120, and 160, and we apply cutout (DeVries & Taylor, 2017) during training (unlike efficient ensembling, which does not use cutout for a fair comparison to prior work). SWE-Multi-Width is a 3 member WRN 28-[7,4,3] ensemble. SWE-Multi-Depth/Width is a 4 member WRN 28-[7,4] 16-[7,4] ensemble. All models start from 4 initial depth-binned SuperWeight Clusters and are refined using the gradient similarity threshold from Eq. (3), $\tau = 0.1$. When learning where to share coefficients, SWE-Multi-Width is trained with the gradient similarity threshold from Eq. (2), $\beta = 0.9$ for CIFAR-100 and $\beta = 0.95$ for CIFAR-10. SWE-Multi-Depth/Width uses $\beta = 0.9$ for CIFAR-100 and $\beta = 0.5$ for CIFAR-10. See Section C.4 for a sensitivity study for hyperparameters $\beta$ and $\tau$.

Note that in Figure 1 of the main paper, we compare against MIMO and BatchEnsemble. These are WRN-28-5 architectures to more fairly compare to our SuperWeight Ensembles for Anytime Inference, which are narrower than WRN-28-10. That said, we also report MIMO and BatchEnsemble results using a WRN-28-10 architecture in the efficient ensembling experiments.

### A.2.1 BASELINES

ShapeShifter Networks (Plummer et al., 2022), Slimmable Networks (Yu et al., 2019), Universally Slimmable Networks (Yu & Huang, 2019b) are each trained using the same training settings as the SWE-Multi-Width and SWE-Multi-Depth/Width models (including using cutout for data augmentation).

**Slimmable Networks** train a Multi-Width network by executing a subset of the channels for a predefined set of width configurations, called switches.

**Universally Slimmable Networks** extend Slimmable Networks to execute any width of a network within a max and minimum by training at each iteration a random subset of switches and calibrating the the batch normalization statistics of the final switches following training. We train Universally Slimmable using a max width of 7, a minimum width of 3, and the number of widths trained at each iteration, $n = 4$. We calculate batch normalization statistics of network widths [3,4,5,6,7] over one epoch following training.

**ShapeShifter Networks (SSNs)** automatically learn where to share parameters within a network through clustering coefficients $\alpha$ to form groups and then within each group generating weights for layers from parameter banks via template mixing methods. For SSNs we give each ensemble member its own independent coefficients $\alpha$ and batch normalization parameters. For the Multi-Depth/Width network the layers are grouped manually by depth. Section 4.1 shows a Multi-Width comparison of our gradient analysis vs. SSN cluster coefficient approach to forming layer groups.

**Multi-Scale Dense Network (MSDN)** (Huang et al., 2018) is a Multi-Depth network that provides dense connectivity between early exits and operates at multiple scales for an efficient Anytime Inference model. We train MSDNet using the training scheme from the original paper. All model hyperparameters are kept the same as the original except the network is built with 16 blocks with 12, 24, and 48 output channels for each of the three scales, respectively, in order to match the inference time with other methods.

**Hierarchical Neural Ensemble** (Ruiz & Verbeek, 2021) is a Multi-Depth network that trains a tree based ensemble using a novel distillation loss. Results are pulled directly from the original paper with models evaluated for inference time using the author's code. The model uses a modified ResNet-50 with $N = 16$ ensemble members.

**Efficient Homogeneous Ensemble Baselines**: BatchEnsemble (Wen et al., 2020) and MIMO (Havasi et al., 2021) are two homogeneous ensembling methods (which we apply to the anytime inference task in Figure 1 of the main paper). BatchEnsemble perturbs weights with rank-1 matrix for each ensemble member. MIMO hard shares all parameters between ensemble members, except for the input convolutional layer and output layer. BatchEnsemble is ineffective as an anytime inference method because there are only $N$ evaluation speeds, where $N$ is the number of ensemble members. MIMO only provides a single inference speed because of its shared backbone.

## B PRIORITY-QUEUE WEIGHT TEMPLATE ASSIGNMENT

In Section 3.2.2 of our main paper we introduce a method that uses gradient similarity between shared parameters in order to determine where sharing is effective. The intuition behind our approach is that sharing between layers is likely less effective when those layers provide conflicting gradients to the shared parameters. We proposed a greedy approach that would train a model for $N$ epochs and then measure similarity between gradients supplied to the shared parameters aggregated over an entire epoch. Gradient similarity between layers $\ell_i, \ell_j$, which we denote as $\psi_{i,j}$, is computed using Eq. (2) when learning where Weight Templates should share coefficients, and Eq. (3) when learning SuperWeight Clusters. This similarity would be used to construct a priority-queue $Q$, which we traverse merging any layers into a group $G$ that are above some threshold $\epsilon$, with any remaining layers being placed into the same group (see Algorithm 1). Note that in our experiments $\epsilon$ is a threshold on gradient similarity, but one could also use a threshold on the number groups instead (or in combination), which we leave for future work. Layers in the same group would continue to share parameters, whereas layers in different groups would no longer completely share parameters. Details on how this is implemented is in Section 3.2.2 for sharing in Weight Templates and Section 3.3 for creating SuperWeight Clusters.

---

**Algorithm 1** Priority-Queue Group Assignment

---

**Input:** threshold $\epsilon$, priority-queue $Q = \{$gradient similarity $\psi_{i,j} : ($layers $\ell_i, \ell_j)\}$

Groups $G = list()$
**while** $|Q| > 0$ **do**
    $\psi_{i,j}, \ell_i, \ell_j = Q.pop()$
    **if** $\psi_{i,j} > \epsilon$ **then**
        **if** $\ell_i \in g$ & $\ell_j \in g'$, where $g, g' \in G$ **then**
            $G.append(g \cup g')$                                   $\triangleright$ Merge into new group
            delete $g, g'$ from $G$
        **else if** $\ell_i \in g$ **then**                      $\triangleright$ One layer already belongs to a group
            $g = g \cup \{\ell_j\}$
        **else if** $\ell_j \in g'$ **then**
            $g' = g' \cup \{\ell_i\}$
        **else**                                 $\triangleright$ Neither layer belongs to a group
            G.append($\{\ell_i, \ell_j\}$)
        **end if**
    **else**                 $\triangleright$ similarity is below threshold, so add any remaining ungrouped layers
        **if** $\ell_i \notin g, \forall g \in G$ **then**
            G.append($\{\ell_i\}$)
        **end if**
        **if** $\ell_j \notin g', \forall g' \in G$ **then**
            G.append($\{\ell_j\}$)
        **end if**
    **end if**
**end while**

    **return** $G$

---

## C   EXPANDED RESULTS

### C.1   SLIMMABLE NETWORKS AND UNIVERSALLY SLIMMABLE NETWORKS ENSEMBLE COMPARISON

Slimmable Networks (Yu et al., 2019) and Universally Slimmable Networks (US) (Yu & Huang, 2019b) both train a dynamic width network such that given a budget at inference time, one can match the budget by running a slim version of the network through executing a subset of the channels at each layer. These methods can be run as an ensemble similar to ours by running inference through multiple widths. In Figure 4 we compare our SWE-Multi-Width model to Slimmable and Universally Slimmable ensembles. SuperWeight Ensembles benefit from ensembling diverse architectures, improving performance. In contrast, ensembling multiple widths of Slimmable and Universally Slimmable Networks only leads to a slight boost, or even decrease, in accuracy.

### C.2   CIFAR-10-C

We present results on CIFAR-10-C below. Compared to efficient ensembling baselines, our SWE-HE outperforms others on two out of three metrics, while also providing flexibility not present in prior work via our heterogeneous ensembles and the ability to adjust the number of parameters in our network without changing architecture.

Table 4: CIFAR-10-C

| Method | Top-1 Acc | NLL | ECE |
|---|---|---|---|
| BatchEnsembles Havasi et al. (2021) | **77.5** | 1.02 | 12.9 |
| MIMO Havasi et al. (2021) | 76.6 | 0.927 | 11.2 |
| SuperWeight Ensembles(**ours**) | 76.01 | 0.885 | 10.2 |
| SuperWeight Ensembles(**ours**) | 76.6 | **0.872** | **8.8** |

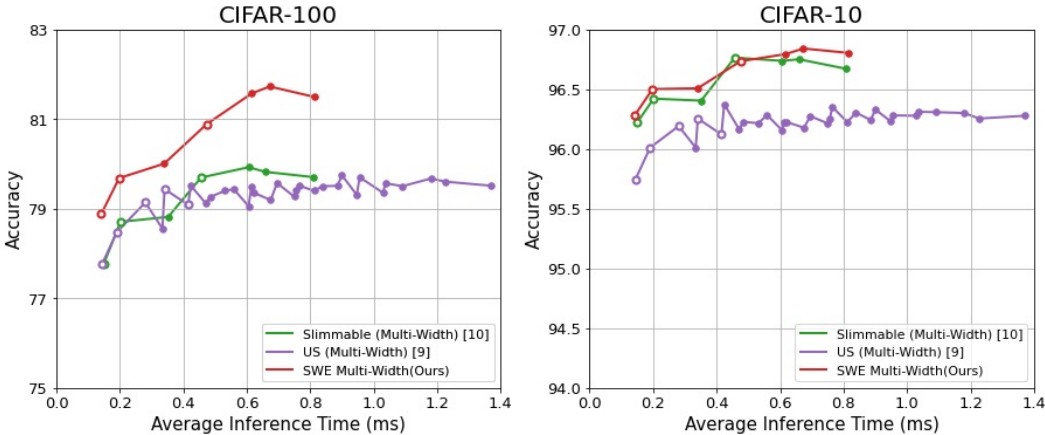

Figure 4: **Slimmable Networks and Universally Slimmable Networks ensemble comparison.** Anytime inference results using Slimmable (Yu et al., 2019) and Universally Slimmable (Yu & Huang, 2019b) as ensembles of multiple widths compared to our SWE-Multi-Width.

## C.3 ENSEMBLE DIVERSITY ANALYSIS

A key attribute of ensembles which makes them effective is their diversity; if the errors of ensemble members are not decorrelated then there is no additional benefit of doing inference through additional ensemble members. In this section, we demonstrate that SuperWeight Ensembles accomplish this.

We use a diversity metric introduced in Fort et al. Fort et al. (2019), which measure the fraction of differing predictions by two ensemble members, normalized the by the error of one of them. We present these results in Table 5; we can see that SuperWeight Ensembles is more diverse than all other shared parameter methods. One interesting finding is that 120M parameter SuperWeight Ensembles outperform Standard Deep Ensembles, yet have lower diversity. Looking deeper, the individual model accuracies are improved for SuperWeight Ensembles (average deep 79.9% vs average SuperWeight Ensembles 80.4%) Therefore, it seems like SuperWeight Ensembles helps the model generalize better. This could come from the fact that the parameter factorization limits the space of possible weights. This is especially interesting because WRN-28-10 is already a highly optimized model from a hyper-parameter perspective. Of note is that one of the ensemble members from the SuperWeight Ensembles gives higher performance (80.5% for the best model) than a standard model (80.1%). Therefore one could use SuperWeight Ensembles to train a highly performant single model, which signifies another advantage our approach has over prior work Havasi et al. (2021); Wen et al. (2020).

To provide another diversity metric, in Figure 5 we interpolate between two WRN-28-10 ensemble members in parameter space to see if the models indeed are in different optimization basins, and report accuracy at each operating point. We accomplish this by interpolating parameters, and leaving Batch Normalization Ioffe & Szegedy (2015) in train mode because accumulated statistics are not meaningful at interpolated points. If the interpolates have high accuracy, this indicates the ensemble members landed in the same optimization basin and therefore are not as diverse as they could be. We find that interpolates have much decreased accuracy compared to the end points, supporting the idea that our model learns diverse ensemble members.

We can see that although all networks experience some degree of accuracy drop between ensemble members, the accuracy drop for the low-parameter model is significantly lower. This seems to indicate that as the function space becomes constrained with a lower number of parameters in the parameter bank, the ensemble diversity starts to suffer.

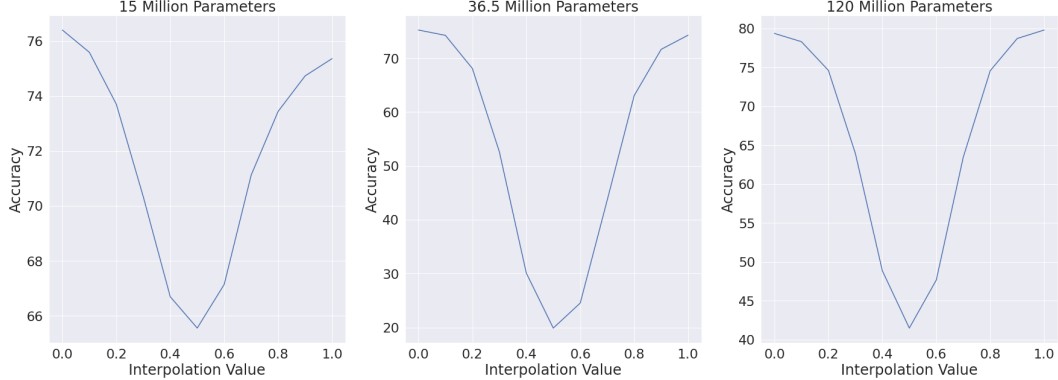

Figure 5: Linear interpolations in parameter space on CIFAR-100, with different numbers of parameters in an ensemble of size 4. We plot accuracy vs interpolation point. Because the accuracy dips, we know the ensemble members are diverse and find different local minima.

Table 5: **Diversity on CIFAR-100.** Diversity is measured as the proportion of samples two ensemble members disagree on, normalized by the error rate of one of the member as in Fort et al. (2019). SuperWeight Ensembles are more diverse than BatchEnsembles, the other shared parameter model. Although our 120M model is slightly less diverse than Standard Deep Ensembles, the performance is higher, due to the average member accuracy being higher than Standard Deep Ensembles

| Method | Params | Diversity |
|---|---|---|
| Standard Deep Ensemble Havasi et al. (2021) | 146.0M | 0.88 |
| BatchEnsembles Havasi et al. (2021) | 36.5M | 0.40 |
| MIMO Havasi et al. (2021) | 36.5M | 0.91 |
| SuperWeight Ensembles(**ours**) | 36.5M | 0.78 |
| SuperWeight Ensembles(**ours**) | 120.0M | 0.85 |

## C.4 GRADIENT ANALYSIS HYPERPARAMETER SENSITIVITY

SuperWeight Ensembles learn where to share parameters through two steps of gradient analysis; first to form SuperWeight Clusters (Section 3.3), and then to separate SuperWeights by decoupling shared coefficients (Section 3.2.2). Here we explore the sensitivity of the hyperparameters used in each step.

When separating SuperWeights using Eq. (2), we give unique coefficients to SuperWeights where the similarity between the gradients of shared coefficients is less than $\beta$. $\beta$ is used as the threshold $\epsilon$ in Algorithm 1 for separating coefficients. In Figure 6(a) we show how sensitive our method is to the selection of $\beta$ using the SWE-Multi-Width architecture on CIFAR-100 (Krizhevsky et al., 2009). In addition to reporting results for values of $\beta \in \{0.1, 0.5, 0.9\}$ (note that $\tau = 0.1$), we also report results when no coefficients are shared, and when all coefficients are shared between layers sharing Weight Templates. The results in Figure 6(a) show that no-coefficient and strict coefficient sharing (referred to as "Not Shared" and "Shared," respectively) underperform compared to using our approach. Note that we found optimal values of $\beta$ to come from dataset-specific tuning.

When generating SuperWeight Clusters using Eq. (3) we split layers into separate groups if the gradient similarity is less than $\tau$. The higher $\tau$, the more groups are formed, the less layers sharing Weight Templates, and the less parameters given to each Weight Template. Setting $\tau \geq 0$ results in sharing between all layers which have gradients that are not conflicting on average. As $\tau$ is increased, layer gradients must point in closer directions to be shared. Note $\tau$ is given as the input value to $\epsilon$ in Algorithm 1. In Figure 6(b) we report best performance comes when $\tau = 0.1$ when using the SWE-Multi-Width architecture on CIFAR-100 (Krizhevsky et al., 2009), which we found to be consistent across settings and datasets. Thus, we use this same value of $\tau$ across all experiments.

## C.5 PERFORMANCE UNDER SEVERE PARAMETER CONSTRAINT

One key feature of SuperWeight Ensembles is that the parameter count is decoupled from backbone. It is therefore interesting to see how the network behaves under stronger parameter constraints. In Table 6, we present CIFAR-100 top-1 accuracies under various parameter constraints. SWE-single superweight refers to the homogenous ensemble which has a single superweight per layer. This is what is presented in Table 1 of our paper. SWE-Gradient Conflict is also a homogenous ensemble, but with learned SuperWeight clusters using the gradient conflict criterion. Finally, SWE heterogenous is a WRN 34-8, 28-12, 28-10, and 28-8 ensemble. The gradient conflict criterion becomes more important at lower parameter counts. The improvement of the heterogenous ensemble over the homogenous ensemble also increases with decreased parameter counts.

Table 6: **Changing parameter constraint:** CIFAR-100 Top-1 accuracies. The gradient conflict criterion becomes more important at lower parameter counts. The improvement of the heterogenous ensemble over the homogenous ensemble also increases with decreased parameter counts.

| Method | 7 million | 15 million | 36.5 million |
|---|---|---|---|
| SWE-Single Superweight | 79.2% | 81.3% | 82.3% |
| SWE - Gradient Conflict | 80.1% | 81.4% | 82.2% |
| SWE- Heterogenous | 80.8% | 81.6% | 82.4% |

## C.6 SUPERWEIGHT CLUSTER MEMBERSHIP

In Figure 7, we present sharing patterns (SuperWeight Cluster membership) of two networks (WRN-28-7/WRN-16-7). The initial depthwise sharing pattern would segment each network into four equal groups of layers. Our gradient-based learning of SuperWeight clusters allows indvidual layers to specialize. Interestingly, it seems like our method learns to separate adjacent layers into separate SuperWeight clusters. This could increase diversity between ensebmle members.

## C.7 DIVERSE ARCHITECTURE FAMILIES

Although we show results primarily on ResNets in our work, our sharing methodology does indeed function well for diverse ensembles. For example, consider an ensemble consisting of a Mobilenetv2 and WRN-28-5, shown in Table C.7. We can see that our sharing procedure even helps compared to standard Deep Ensemble performance, despite diverse architectures. Note that these results are over 2 ensemble members, whereas the results in Table 1 of our paper use 4 ensemble members.

## C.8 EFFICIENT ENSEMBLES ON CIFAR

In Table 8 we present CIFAR results from the main paper, with error bars (representing standard error) to provide additional context. Note that even with this additional information, it is clear our method outperforms all baselines on CIFAR in the low parameter regime, and even outperforms standard ensembles in the high parameter regime (with 17% fewer parameters). Results we report for baselines are taken from prior work and do not provide error bars. See Section 4 of the main paper for more discussion.

Table 7: **Diverse Architechture Families:** Here we show an ensemble consisting of a Mobilenetv2 and WRN-28-5, shown in Table. We can see that our sharing procedure even helps compared to standard DeepEnsemble performance, despite diverse architectures.

| Method | Top-1 | NLL | ECE |
|---|---|---|---|
| Deep Ensembles | 80.1% | 0.776 | 6.5% |
| SWE-HE | 80.3% | 0.762 | 5.33% |

Table 8: **Homogeneous ensembling comparison** on CIFAR-100 (clean) (Krizhevsky et al., 2009) CIFAR-100-C (corrupt) (Hendrycks & Dietterich, 2019), and CIFAR-10 (Krizhevsky et al., 2009) using the WideResNet architecture (Zagoruyko & Komodakis, 2016) averaged over three runs. We add error bars representing standard error for our experiments for context. Prior work did not report error bars. **(a)** shows that our approach outperforms prior work in efficient ensembling. **(b)** compares the performance of increasing the number of parameters (without changing the architecture) using our approach compared to standard Deep Ensembles, which trains 4 independent networks as ensemble members. See Appendix C.8 and Section 4 of the main paper for more discussion.

| | | | CIFAR-100 (clean) | | | CIFAR-100-C | | | CIFAR-10 | |
|---|---|---|---|---|---|---|---|---|---|---|
| Method | Params | Top-1 ↑ | NLL ↓ | ECE ↓ | Top-1 ↑ | NLL ↓ | ECE ↓ | Top-1 ↑ | NLL ↓ | ECE ↓ |
| **(a)** WRN-28-10 | 36.5M | 79.8 | 0.875 | 8.6 | 51.4 | 2.70 | 23.9 | 96.0 | 0.159 | 2.3 |
| BE | 36.5M | 81.5 | 0.740 | 5.6 | 54.1 | 2.49 | 19.1 | 96.2 | 0.143 | 2.1 |
| BE + EnsBN | 36.5M | 81.9 | n/a | 2.8 | 54.1 | n/a | 19.1 | 96.2 | n/a | 1.8 |
| MIMO | 36.5M | 82.0 | 0.690 | 2.2 | 53.7 | 2.28 | 12.9 | **96.4** | 0.123 | 1.0 |
| SWE-HO (**Ours**) | 36.5M | 82.2 ± 0.28 | 0.702 ± 0.009 | 2.7 ± 0.04 | 52.9 ± 0.24 | **2.17 ± 0.02** | 10.3 ± 0.47 | 96.3 ± 0.05 | 0.120 ± 0.002 | **0.8 ± 0.08** |
| SWE-HE (**Ours**) | 36.5M | **82.4 ± 0.02** | **0.663 ±0.001** | 3.0 ± 0.23 | 53.0 ± 0.06 | **2.17 ± 0.01** | 10.0 ± 0.45 | **96.5 ± 0.02** | **0.115 ± 0.003** | **0.8 ± 0.03** |
| **(b)** Deep Ensembles | 146M | 82.7 | **0.666** | **2.1** | 54.1 | 2.27 | 13.8 | **96.6** | **0.114** | 1.0 |
| SWE-HO (**Ours**) | 120M | **82.9 ± 0.05** | **0.666 ± 0.007** | 2.2 ± 0.3 | **54.7 ± 0.05** | **2.00 ± 0.01** | **10.3 ± 0.01** | **96.6 ± 0.09** | 0.119 ± 0.001 | **0.8 ± 0.07** |

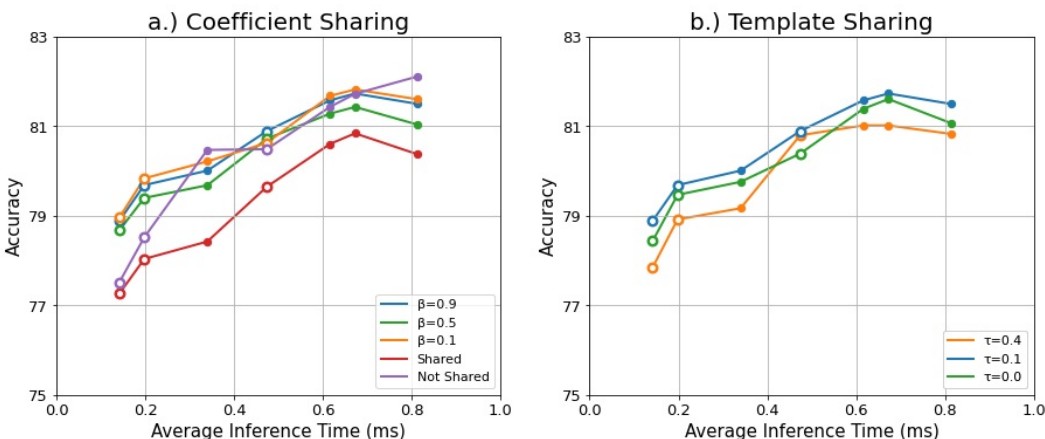

Figure 6: **Gradient analysis hyperparameter sensitivity.** Anytime inference results on CIFAR-100 (Krizhevsky et al., 2009) using SWE-Multi-Width architecture when: **a)** varying the gradient similarity threshold $\beta$ from Eq. (2) for coefficient sharing, no coefficient sharing, and strict coefficient sharing; and **b)** varying the gradient similarity threshold $\tau$ from Eq. (3) for SuperWeight Cluster forming. For both $\beta$ and $\tau$ best performance is between 0.0 and 1.0, indicating that too much sharing and too little sharing are both harmful.

Figure 7: **SuperWeight Cluster Membership** of a two member ensemble, a WRN-28-7 and WRN-16-7. The initial depthwise sharing pattern would segment each network into four equal groups of layers. Our gradient-based learning of SuperWeight clusters allows indvidual layers to specialize. Interestingly, it seems like our method learns to separate adjacent layers into separate SuperWeight clusters. This could increase diversity between ensemble members.

