# OpenReview forum: "SuperWeight Ensembles: Automated Compositional Parameter Sharing Across Diverse Architechtures"
_ICLR.cc/2023/Conference — Submitted to ICLR 2023_

### Official Review · Reviewer_owxX · 2022-10-24

**Confidence:** 2
**Correctness:** 4
**Technical Novelty And Significance:** 3
**Empirical Novelty And Significance:** 3
**Recommendation:** 6

**Clarity, Quality, Novelty And Reproducibility:**

- **Clarity:** The method is understandable. However, the motivation and the problem that the current paper is tackling compare to the previous works seem quite unclear to me. It is understandable but I hope the authors clarify what will be the main limitation of previous works and what is the difference and contributions of their approaches.
- **Quality:** This paper shows a good quality presentation of their methods and results.
- **Novelty:** The idea is very novel to me because this approach could generate parameters for both homogeneous networks and heterogeneous networks. However, the criteria, such as using gradient similarity seem to lack novelty.
- **Reproducibility:** I think this paper shows quite a low reproducibility. It is difficult for me to understand the very details of each step of the approach. For example, how to set weight templates is unknown. How to learn \alpha is also unknown.

**Strength And Weaknesses:**

**Strength**

- This paper propose parameter sharing from SuperWeight for both homogeneous architectures and heterogeneous architectures.
- The proposed method is able to obtain better performance both in ensembling and in any time inference tasks while it shows a marginal improvement in ensembling compared to baselines.
- The approach that the current paper is proposing seems reasonable and convincing.

**Weakness**

- Since the proposed methods contain several steps to construct diverse architectures with the sharing parameters, it was difficult to understand at first for me. I hope the authors provide a broad concept for each step which is more abstract than figure 2.
- As mentioned in the related works, I think the author also should compare the performance with the zero-shot NAS which does not cost at all to find high-performance networks.
- There are missing results of CIFAR-10-C and ImageNet-C. I hope the authors to provide the experimental results also in CIFAR-10-C and ImageNet-C.

**Question**

I might have missed the following information in the paper.
- How to generate templates is unclear in section 3.1. Is \alpha learnable variables? Or deterministic variables?
- How to construct each member from the layer weight?
- What is the stop criteria that is explained in the 4. of Section 3?

**Summary Of The Paper:**

This paper proposes SuperWeight Ensembles which generate parameters of diverse architectures from single SuperWeight parameters. To efficiently find the parameters, authors suggest making SuperWeight from templates with the linear combination using coefficients \alpha. Then, cluster the weights based on the gradient similarity. After constructing SuperWeight, layer weight construction determines the coefficients for generating a layer-wise SuperWeight and assembled into diverse members of architecture. SuperWeight Ensembles are able to generate parameters for both homogeneous architectures and heterogeneous architectures.

**Summary Of The Review:**

Overall, I recommend marginally above the acceptance threshold. I hope the authors could resolve my concerns in the rebuttal.

---

### Official Review · Reviewer_Qjhr · 2022-10-24

**Confidence:** 3
**Clarity, Quality, Novelty And Reproducibility:** The paper could benefit from a cleare…
**Correctness:** 3
**Technical Novelty And Significance:** 2
**Empirical Novelty And Significance:** 2
**Recommendation:** 5

**Strength And Weaknesses:**

- The paper proposes a new way to form deep ensembles which is not seen in the literature
- The paper could benefit from an explanation of the results


Open questions and comments
- From the paper, the reader is required to go over [1] to understand the template generating process, as this is not a standard approach I would suggest to devote a section of the paper to this part, which in turn would also make the paper accessible to a wider audience.
- It woul be ideal if the authors could outline the contributions of the paper vs [1].
- In Fig 1 caption the authors claim their approach allows them to learn optimal soft parameter sharing. Why is it optimal?
- Most results don’t show confidence intervals.
- Table 2. What would happen if the authors added an ablation study for different # of parameters for imagenet? Does adding more parameters match the performance level for deep ensembles or perform even better?
- Table 1(b) is the text is referred as table 1(c).
- The paper could also benefit from a discussion of the computational implications/limitations of the proposed approach during training time.  This would be informative for readers considering to use the proposed approach over standard deep ensembles.
- The authors mention that because they need to cluster layers across models by layer similarity they use a coefficient lambda to check the gradient cosine similarity. In this case, the layers would need to have the same form don’t they? Could be combine layers from a vit model with a can model using this approach? The authors cite Raghu which notes that there are several differences across the representations, so this part is confusing to me.

**Summary Of The Paper:**

This paper builds from the work in [1] to make efficient ensembles. This paper proposes to build ensembles by using shared templates/layers across ensemble members. Unlike previous work in this domain, they mention that because they can use average weights of different dimensions [1], they can combine models with layers of different sizes which allows them to form ensembles of different sizes given fixed parameter setting, which can help tradeoff accuracy vs compute time.
[1] Plummer et al. 2022


**Summary Of The Review:**

Overall this paper proposes an interesting method motivated by an important problem (anytime prediction) not usually motivated in the literature. The paper would greatly benefit from a more detailed discussion of the methods and the results (see above) to improve its accessibility to a wider audience.

---

### Official Review · Reviewer_d45p · 2022-10-25

**Confidence:** 2
**Correctness:** 3
**Technical Novelty And Significance:** 2
**Empirical Novelty And Significance:** 3
**Recommendation:** 5

**Clarity, Quality, Novelty And Reproducibility:**

Overall the clarity is good, but the writing of algorithm part needs to be polished. The experiments seems to be reproducible.

**Strength And Weaknesses:**

The proposed method is flexible, which allows ensembles of network with different architectures, and enables anytime inference, which is important.

Better performance with fewer parameters are obtained, which illustrates the effectiveness of the proposed method.

Writing of section 3 needs to be polished, currently it is a little bit vague.

Some ablation studies are suggested, how does each component/step of the algorithm contribute to the performance. For example, what will the result be if we only have one cluster, or only have one group in each cluster?

How about the training time needed compared to other related works.

How will the performance be on more complicated datasets other than CIFAR.



**Summary Of The Paper:**

The paper propose SuperWeight Ensemble, a new method for parameter sharing and network ensemble which is more flexible. Better results are obtained in experiments.

**Summary Of The Review:**

The paper propose a useful method which leads to flexible and effective parameter sharing and ensemble, good experimental results are obtained.

---

### Decision · Program_Chairs · 2023-01-20

**Decision:**

Reject

**Justification For Why Not Higher Score:**

The construction of specific network weights is a bit unclear even for the reviewer that has positive support. There were questions about time for training and more complex datasets. There were also questions of the relationship to Plummer 2022. There was no author rebuttal and the reviewers on average are negative.

**Justification For Why Not Lower Score:**

N/A

**Metareview: Summary, Strengths And Weaknesses:**

This paper is about an efficient way to share parameters across an ensemble in a way that allows for members of the ensemble to be different, with the goal of reducing the memory requirements. The method supports anytime predictions. At a high level, this approach takes weight templates and assembles the superweights. The general idea is interesting and has potential for leading to diverse, efficient ensembles. Prior to the author's review response, there were questions about 1) the relationship to Plummer 2022, 2) time for training and more complex datasets, and 3) clarity of the actual method. In the author's response, the relationship to Plummer was made clear. More experiments were also reported, though some of them such as Imagenet-C asked for by the positive reviewer, have middling results. However, the issue of clarity remains and is a large one. For example, after reading the author response, the one reviewer in favor of acceptance writes

"However, I think the current work needs to be more polished since all reviewers are having difficulties understanding the process of the work and the difference between the previous work. Moreover, I also agreed with the concerns that are raised by other reviewers. Therefore, I will decrease confidence to 2."